# Melatonin Regulates the Daily Levels of Plasma Amino Acids, Acylcarnitines, Biogenic Amines, Sphingomyelins, and Hexoses in a Xenograft Model of Triple Negative Breast Cancer

**DOI:** 10.3390/ijms23169105

**Published:** 2022-08-14

**Authors:** Rubens Paula Junior, Luiz Gustavo de Almeida Chuffa, Vinicius Augusto Simão, Nathália Martins Sonehara, Roger Chammas, Russel J. Reiter, Debora Aparecida Pires de Campos Zuccari

**Affiliations:** 1Cancer Molecular Research Laboratory, Department of Molecular Biology, Faculty of Medicine of the São José do Rio Preto, São Paulo 15090-000, Brazil; 2Department of Structural and Functional Biology, Institute of Bioscience of Botucatu, Botucatu 18618-689, Brazil; 3Center for Translational Research in Oncology, Institute of Cancer of São Paulo State, University of São Paulo, São Paulo 01246-000, Brazil; 4Department of Cell Systems and Anatomy, UT Health, San Antonio, TX 78229, USA

**Keywords:** breast cancer, melatonin, metabolomics, xenografted mice, plasma metabolites, circadian profile

## Abstract

Metabolic dysregulation as a reflection of specific metabolite production and its utilization is a common feature of many human neoplasms. Melatonin, an indoleamine that is highly available during darkness, has a variety of metabolic functions in solid tumors. Because plasma metabolites undergo circadian changes, we investigated the role of melatonin on the profile of amino acids (AAs), biogenic amines, carnitines, sphingolipids, and hexoses present in the plasma of mice bearing xenograft triple negative breast cancer (MDA-MB-231 cells) over 24 h. Plasma concentrations of nine AAs were reduced by melatonin, especially during the light phase, with a profile closer to that of non-breast cancer (BC) animals. With respect to acylcarnitine levels, melatonin reduced 12 out of 24 molecules in BC-bearing animals compared to their controls, especially at 06:00 h and 15:00 h. Importantly, melatonin reduced the concentrations of asymmetric dimethylarginine, carnosine, histamine, kynurenine, methionine sulfoxide, putrescine, spermidine, spermine, and symmetric dimethylarginine, which are associated with the BC metabolite sets. Melatonin also led to reduced levels of sphingomyelins and hexoses, which showed distinct daily variations over 24 h. These results highlight the role of melatonin in controlling the levels of plasma metabolites in human BC xenografts, which may impact cancer bioenergetics, in addition to emphasizing the need for a more accurate examination of its metabolomic changes at different time points.

## 1. Introduction

Breast cancer (BC) is the most frequent malignancy in females, and is highly heterogeneous at the molecular level [1]. There are many classifications of BC, and clinicians usually identify their subtypes based on histological and molecular characteristics [2]. Among BC subtypes, triple-negative breast cancer (TNBC) has a particular molecular profile which is often aggressive and lacks targeted therapies [3]. Recently, a multi-omics study classified TNBC samples according to three metabolic features: the lipogenic subtype (1) showing increased lipid metabolism; the glycolytic subtype (2) with enhanced carbohydrate and nucleotide metabolism; and the mixed subtype (3) with partial metabolic dysregulation [4]. Recognizing the daily metabolic signatures of tumors may favor the development of reprogramming strategies or personalized treatments. 

Metabolomics, an omics science for the study, identification, and quantification of metabolites (small molecules < 1500 Da), is considered a high throughput technology for mapping new candidates and altered metabolic networks in cancer [5]. Cancer cells dynamically change their energy metabolism, and various metabolic intermediates accumulate in tumors [6]. Notably, the presence of certain metabolites associated with variations in their concentrations may indicate predisposing factors or even a particular disease [7]. Although deregulated uptake of glucose and amino acids and utilization of their intermediates in the glycolysis/tricarboxylic acid (TCA) cycle have been listed as the main contributors to tumorigenesis [8], shifts involving lipids (e.g., sphingolipids), acylcarnitines, hexoses, and biogenic amines have gained attention in the context of cancer cell metabolism [9]. Metabolic dysregulation is a hallmark of cancer, and is a direct and indirect consequence of oncogenic mutations [10]. Plasma metabolite levels can be associated with patient survival in addition to serving as tumor biomarkers [11]. Identifying metabolites in fluids (e.g., blood, urine) is more advantageous and promising because it requires easy and non- or minimally-invasive collection [12].

Studies in cancer cells have revealed numerous metabolic pathways involved in energetic and biosynthetic demands which are required for cell growth and proliferation. For instance, the levels of many amino acids are higher in different cancer types than in normal tissue [13]. Owing to its high energy demand, the carnitine shuttle system is active for fatty acyl transportation during fatty acid β-oxidation in tumor cells [14]. Likewise, bioactive sphingolipids and biogenic amines play important roles in cancer cell growth and death, thereby becoming targets for anti-cancer therapies [15,16]. To guide preventive actions for BC, a wide range of metabolites, including acylcarnitines, amino acids, biogenic amines, glycerophospholipids, sphingolipids, and hexoses, were correlated with mammographic density (MD), which is considered a stronger risk factor for young women [17]. Interestingly, sphingomyelin (SM) C16:1 and phosphatidylcholine (PC) C30:2 showed an inverse association with MD percentage.

Generally, cellular metabolism, including physiological and molecular events, shows a diurnal rhythm pattern governed by circadian clocks located in central and peripheral tissues as well as by external factors such as light, meals, and social timing [18]. In humans and mice, metabolic studies have documented broad variation in blood molecules (e.g., amino acids, acylcarnitines, carbohydrates, fatty acids, and phospholipids) under clock-dependent oscillation [19,20]. Because melatonin is secreted in a circadian manner and possesses several antitumor actions [21,22], we hypothesized that melatonin could orchestrate BC cell metabolism throughout the 24-h period. Melatonin is an indoleamine produced by the mitochondria of perhaps all animal cells [23,24], reaching its highest levels in this specific cell compartment. From a metabolomic standpoint, melatonin alters glucose metabolism, TCA metabolites, and ATP production in prostate cancer cells [25]. A recent study using metabolomic analysis documented that exogenous melatonin regulates metabolic pathways in head and neck squamous cell carcinoma [26].

The circadian profile of the tumor microenvironment may have a clinically significant value, and the day and night times may be determinants of the rhythmic circadian production of many metabolites. The circadian rhythm and concentrations of melatonin are disturbed in BC patients [27], thus affecting tumor aggressiveness and quality of life. As clinical studies have shown an inverse correlation between melatonin metabolites and the risk of BC [28], and since melatonin is a beneficial and minimally toxic agent, the current study investigated plasma metabolite concentrations in a breast cancer xenograft mouse model throughout a 24 h period after treatment with melatonin.

## 2. Results

### 2.1. Plasma Concentrations of Amino Acids Varied during Day and Night Time in Breast Cancer Mice Treated with Melatonin

The profiles of 21 amino acids (AAs) were first analyzed during the day and night over 24 h by comparing tumor and non-BC-bearing animals after treatment with 40 mg/kg of melatonin over a total of eight time points. We identified a distinct profile of AAs in which their levels were higher at night than during the day. More specifically, the tumor group presented elevated levels of AAs during the day and was distinctively clustered with the other groups. In contrast, at night, the tumor group showed lower plasma levels of AAs compared to the non-tumor and melatonin-treated tumor-bearing animals, which presented a more similar profile (Figure 1A). Four independent clusters (alanine, glutamine, lysine-glycine-valine, and other AAs) were observed among AAs in the experimental groups. In the BC xenograft model, melatonin treatment resulted in AA concentrations close to those in the non-tumor mice, suggesting that melatonin influenced their daily production and consumption. Using Pearson’s correlation coefficient, we examined the correlation among the different AA levels in the experimental groups considering their daily production. Notably, glycine and lysine were negatively correlated with several AAs, whereas arginine, aspartate, citrulline, glutamate, glutamine, histidine, isoleucine, leucine, methionine, ornithine, phenylalanine, proline, threonine, tryptophan, and valine were positively correlated (Figure 1B).

To effectively specify the daytime variation of AA concentrations, we further evaluated four time points during the day (06:00 h, 09:00 h, 12:00 h, and 15:00 h, 21:00 h, 00:00 h, and 03:00 h). Notably, melatonin treatment significantly reduced the levels of arginine, citrulline, leucine, lysine, methionine, ornithine, and proline at 06:00 h and 09:00 h (Figure 2). Melatonin also reduced aspartate (09:00 h and 12:00 h), tryptophan (15:00 h), and tyrosine (06:00 h) levels. Conversely, BC-bearing mice treated with melatonin showed significantly increased levels of aspartate (18:00 h), leucine (00:00 h and 18:00 h), lysine (15:00 h), proline (00:00 h and 21:00 h), serine (15:00 h), and valine (21:00 h), which were similar to the levels detected in non-tumor mice treated with melatonin (Figure 2). The other AAs did not exhibit daily variation after treatment (data not shown).

To broadly investigate the effects of melatonin on the daily AA concentration, we calculated the mean differences of each AA and performed analysis with a total of 12 metabolites over the day and night over a period of 24 h after 21 days of treatment. Interestingly, at night, melatonin enhanced the plasma levels of valine, proline, ornithine, methionine, lysine, leucine, isoleucine, histidine, citrulline, and asparagine compared to the levels in control BC animals, whereas it significantly reduced tryptophan, proline, ornithine, methionine, lysine, isoleucine, glutamate, and citrulline during the day in BC-bearing mice (Figure 3). Upon melatonin treatment, daytime reduction of these AAs, which are essential for tumor progression, may affect BC cell metabolism, resulting in less aggressive behavior.

### 2.2. Acylcarnitines Are Significantly Altered by Melatonin in BC-Bearing Mice

The carnitine system is considered a gridlock that promotes metabolic flexibility of tumor cells [29]. In this context, plasma levels of acylcarnitines were analyzed in tumor and non-tumor bearing animals treated or untreated with melatonin. Among the 24 acylcarnitines, melatonin reduced 12 molecules in BC animals compared with their controls. Specifically, melatonin treatment significantly reduced decenoylcarnitine and dodecenoylcarnitine levels at 06:00 h and 15:00 h compared to the control animals (Figure 4). Moreover, melatonin reduced the levels of hexadecadienylcarnitine, hexadecanoylcarnitine, hexadecenoylcarnitine, hydroxyhexadecenoylcarnitine, hydroxyoctadecenoylcarnitine, hydroxytetradecenoylcarnitine, octadecadienylcarnitine, tetradecanoylcarnitine, tetradecadienylcarnitine, and tetradecenoylcarnitine at the 15:00 h (Figure 4).

### 2.3. Melatonin Reduces the Levels of Biogenic Amines in BC-Bearing Mice 

Biogenic amines are well-known precursors to carcinogenic N-nitroso compounds. To explore the influence of melatonin on these molecules, we evaluated its circadian levels throughout the day and night. As depicted in Figure 5, melatonin significantly reduced the concentrations of asymmetric dimethylarginine (03:00 h), carnosine and histamine (15:00 h), kynurenine (12:00 h and 15:00 h), methionine sulfoxide (00:00 h, 03:00 h, 06:00 h, and 09:00 h), putrescine (09:00 h and 15:00 h), spermidine (15:00 h and 18:00 h), spermine (09:00 h and 12:00 h), and symmetric dimethylarginine (06:00 h and 15:00 h). Importantly, these biogenic amines are associated with the enrichment of metabolic sets including breast cancer, Alzheimer’s disease, histidinemia, polycystic kidney disease, nephrotic syndrome, and stroke. 

### 2.4. Melatonin Reduces the Levels of Sphingolipids in BC-Bearing Mice over Specific Time Points

Sphingolipids function as bioactive molecules responsible for the regulation of cancer cell signaling by controlling tumor suppression and survival. The metabolic network of sphingomyelins (SM) was examined in the plasma of BC animals that received melatonin treatment. As shown in Figure 6, a remarkable reduction in SM concentration ((OH) C14:1, (OH) C16:1, (OH) C22:1, (OH) C22:2, (OH) C24:1, C24:0, C24:1, C26:0, and C26:1) occurred at 06:00 h after melatonin treatment. In general, two different profiles were observed varying their levels over melatonin and vehicle-treated tumors. Melatonin treatment formed a cluster with reduced mean levels of SM, especially at 00:00 h, 09:00 h, 12:00 h, 15:00 h, and 18:00 h compared to the high levels observed in control animals at 09:00 h, 15:00 h, and 18:00 h (Figure 6). 

### 2.5. Concentration of Hexoses Is Significantly Reduced by Melatonin in BC-Bearing Mice

The self-sufficiency of tumor cell bioenergetics requires hexose sources, especially those from glucose. Concentrations of hexoses (including glucose) were altered by melatonin at specific time points. Hence, plasma hexoses were significantly reduced by melatonin at 15:00 h, 18:00 h, and 21:00 h when compared to BC-bearing animals that received the vehicle only (Figure 7A). By examining the concentration of hexoses after melatonin treatment, we observed a significant reduction over the entire light phase (15.3% lower than control animals) and dark phase (20.7% lower than control) (Figure 7B), which may reflect its anti-glycolytic action in cancer cells.

### 2.6. Top Enrichment Analysis Based on the Metabolites Modified by Melatonin

We evaluated all metabolites in the samples of BC-bearing mice. KEGG analysis revealed that the main metabolites altered by melatonin treatment enriched important cell signaling pathways, including aminoacyl-tRNA biosynthesis, arginine and proline metabolism, beta-alanine, glutathione, and histidine metabolism, valine, leucine and isoleucine metabolism, and galactose and tryptophan metabolism, among others (Figure 8A). Based on the chemical class of metabolites, we observed that most are organic acids, followed by organic nitrogen compounds, carbohydrates, organic oxygen compounds, and sphingolipids (Figure 8B).

## 3. Discussion

Herein, we report a clear difference in the secretion of plasma metabolites between an in vivo model of BC xenograft mice and non-tumor-bearing mice throughout a 24 h light:dark period (Figure 9). When analyzing normal and BC-bearing animals, there were no rhythmic variations in plasma metabolites over 24 h; on the contrary, we observed a specific time point dependence, mainly in the BC animals treated with melatonin. Overall, melatonin led to a reduction in the availability of plasma AAs, acylcarnitines, sphingomyelins, biogenic amines, and hexoses in BC-bearing mice, especially during the daytime (light phase).

Plasma samples were collected as a reliable source of tumor biomarkers. Tumor-bearing mice treated with the vehicle showed peak levels of 12 AAs, most of which were enhanced in the middle light phase and reduced in the early dark phase. The reason for this observation is not clear, and may be related to the differential expression of clock gene transcription/translation which in turn controls physiological oscillations. In fact, the expression of certain specific circadian genes (*BMAL1*, *PER2*, and *TIMELESS*) has been demonstrated to be disrupted in breast cancer cell lines [30,31,32,33,34]. Another possible explanation is that BC originating from MDA-MB-231 cells disturbs liver metabolism, resulting in altered plasma AAs [35].

Cancer cells require higher amounts of AAs to satisfy the metabolic demands associated with their proliferation (e.g., for protein and DNA synthesis), which leads to reduced availability of AAs in the plasma [36]. In the current study, the concentrations of AAs were decreased in the BC xenograft animals treated with melatonin in the early light phase (06:00 h and 09:00 h) to values near those of the non-BC-bearing mice. It is known that melatonin directly suppresses the metabolic and proliferative activities of breast cancer xenografts [37,38]. This feature was shown by Blask and co-workers [37], who reported the suppression of proliferative and metabolic activities in human BC xenografts acutely perfused with blood collected during the mid-dark phase, which was abrogated after exposure to dim light at night. According to Blask et al. [38], there is a balance between nocturnal circadian melatonin signaling and tumor growth dynamics characterized by an up-regulation in daytime cell metabolism, signaling, proliferation, and survival, which is offset by a substantial downregulation of these activities by melatonin during the night. This suggests that the lower levels of some AAs observed in the BC-bearing mice treated with melatonin are a consequence of the circadian host–cancer balance being disrupted by the increased levels of melatonin after the treatment period, which may influence daily AA production and consumption.

Despite the low availability of AAs, our xenograft model did not show changes in body mass throughout the experiments, and tumor growth was confined to the subcutaneous layer. It has been reported that xenograft tumors in the fat pad require more time for consistent growth than those in the hind flank [39,40,41], in addition to accounting for reduced tumor volumes. A reduction in tumor volume is expected after melatonin treatment; however, the suppressive effects of melatonin on the proliferation of MDA-MB-231 cells were inconclusive in our model. Nonetheless, studies from our group have previously shown the effectiveness of melatonin on MDA-MB-231 cells in different experimental conditions [22,39]. Jardim-Perassi et al. [39] showed that a reduction in cell viability is only achieved with pharmacological concentrations of melatonin (1 mM). In vivo analysis also showed that melatonin significantly reduced tumor volume, with the tumor growth occurred in the hind flank. The effects of melatonin on the inhibition of estrogen-mediated proliferation of human ER-α positive BC cell lines (e.g., MCF-7) are well known [42]; however, despite several studies showing an anti-proliferative effect of melatonin on the triple-negative MDA-MB-231 cell line [43,44], others have reported that melatonin did not significantly inhibit in vitro proliferation of MDA-MB-231 cells [45,46]. One possible mechanism is that the effectiveness of melatonin may be dependent on the expression of the estrogen receptor [47]. These differences remain to be elucidated, but may explain, at least in part, the absence of alteration in tumor mass at the end of melatonin treatment in the xenograft model. More recently, studies focused on combination therapy using melatonin have documented a potent inhibitory action on MDA-MB-231 cells. Hasan et al. [48] reported that melatonin, estrogen, and progesterone led to significant inhibition of cell proliferation. In a HER2-positive BC mice model, these compounds delayed tumor onset while lowering its incidence. In addition, the same group showed that melatonin–tamoxifen combination is capable of promoting higher affinity to ESR1 and MT1R, which increases potency and efficacy to inhibit migration versus cell viability in a patient-derived xenograft TNBC cell line (TU-BcX-4IC) [49]. 

We observed that certain AAs from BC xenografts increased or decreased to basal levels similar to those of non-tumor-bearing animals depending on the time of day. The tumor-free animals had higher bioavailability of certain plasma AAs, especially at 12:00 h, which is likely due to a particular diet metabolism over the day. Levels of AAs such as arginine, tryptophan, tyrosine, leucine, lysine, ornithine, citrulline, methionine, aspartate, and proline were significantly reduced in the early light phase of the day in the plasma of BC-bearing mice treated with melatonin, while lysine and serine levels were increased in the late light phase of the day. Aspartate, proline, leucine, and valine also exhibited an increased value close to that of the non-tumor mice treated with melatonin in the early dark phase of the day. While variations in AAs metabolism and its concentration in human BC patients have been extensively investigated [17,36,50,51,52], our study is the first to demonstrate the effects of melatonin on the AAs profile in a BC xenograft model. 

According to Eniu and colleagues [50], arginine, alanine, isoleucine, tyrosine, and tryptophan are suitable biomarkers for the diagnosis and prediction of BC progression. These AAs present a progressive decline between stages I and III of BC, with alanine and arginine being capable of stimulating cell proliferation [36,50]. In our study, enrichment analysis showed that among the main metabolites altered by melatonin are valine, leucine, isoleucine, and arginine biosynthesis; thus, the daytime reduction of these AAs may affect cancer cell metabolism, possibly resulting in less aggressive behavior, as they are essential for BC progression. Notably, MDA-MB-231 cells are classified as an arginine auxotrophic cell line [53], which cannot synthesize arginine owing to low levels of argininosuccinate synthase [54]. In this particular case, a reduction in circulating arginine by melatonin would be harmful for this BC-derived cell line, as these cells are unable to compensate for the production of this metabolite.

The altered profile of AAs represents some of the deviations in the metabolic activity associated with BC, and several other pathways such as the carnitine metabolism pathway (i.e., acylcarnitines and related enzymes) have been shown to be affected in the BC condition [17,52,55,56]. Acylcarnitines are key intermediates in the β-oxidation of fatty acids in the mitochondria and regulate gluconeogenesis [29,57], intertwining crucial metabolites, factors, and pathways to provide fundamental bioenergetic sources for cancer cell survival [58]. We initially demonstrated that melatonin treatment significantly reduced the concentrations of several medium- and long-chain acylcarnitines, especially at 15:00 h. Levels of tetradecenoylcarnitine (C14:1) [17,52], tetradecadienylcarnitine (C14:2) [52], hexadecanoylcarnitine (C16), hexadecenoylcarnitine (C16:1) [52,56], hexadecadienylcarnitine (C16:2) [56], and octadecadienylcarnitine (C18:2) [55], which were changed by melatonin treatment, are often elevated in both human and xenograft BC models.

In tumor-bearing animals, melatonin decreases the concentrations of several biogenic amines, mainly during the day. Biogenic amines are endogenous compounds involved in intercellular communication, membrane stability, oxidative stress, and regulation of cell growth [59]. Although we observed a reduction in kynurenine and symmetric dimethylarginine (SDMA) in the TM group compared with the TV group during the day, His et al. [17] did not observe a positive correlation between these biogenic amines and BC risk. Interestingly, kynurenine was shown to promote several changes in BC cells, including the backward conversion of melatonin to N-acetylserotonin in the mitochondria, which promotes the activation of tyrosine receptor kinase β, ultimately increasing the survival and migration of BC cells [60]. Curiously, plasma levels of melatonin were shown to be significantly higher in patients with metastatic BC disease than in those with controlled or stable conditions [61], which is possibly related to indirect stimulus for melatonin production by gut microbiota dysbiosis, which activates the kynurenine pathway for melatonin consumption by BC cells [62]. As our tumor-bearing animals did not develop metastasis, we believe that early treatment with melatonin during the establishment of BC may act to control disease aggressiveness, and therefore lower the kynurenine status.

Polyamines such as putrescine, spermidine, and spermine are arginine/ornithine derivatives with great importance because of their essential role in maintaining macromolecular synthesis and cell proliferation rates [59]. They are frequently dysregulated, and are present in higher concentrations in tumor cells and growing tissues [63]. However, studies on the metabolic interplay between polyamines and BC are scarce. Herein, we show for the first time that BC xenografts treated with melatonin had the lowest plasma concentrations of putrescine, spermidine, and spermine at different times of the day; these polyamines have been previously detected at higher levels in saliva [64,65] and urine samples [66] of BC subjects. It should be considered that both polyamines and biogenic amines can bind covalently with other molecules, promoting protein modifications mostly related to the regulation of cell death, mobility, and invasion [59].

In tumor-bearing mice, melatonin treatment reduced the mean sphingomyelin levels, especially during the day, despite a remarkable increase observed at 12:00 h. Sphingomyelins are a class of sphingolipids implicated in the maintenance of lipid membrane structure and are synthesized from ceramide by sphingomyelin synthase 2, which promotes BC development by regulating cell proliferation, migration, and invasion in patients with metastatic BC [67]. High concentrations of sphingomyelins in the plasma of BC subjects were observed by Qiu and colleagues [55], including (OH) C14:1, (OH) C16:1, (OH) 22:1, (OH) 22:2, (OH) 24:1, and C26:0; all of which were diminished in the light phase of the day in BC xenografts after melatonin treatment. These effects of melatonin on the metabolic profile of sphingomyelins and other circulating metabolites may be related to its prospective effects during daytime; as previously reported by Cipolla-Neto and Amaral [68], melatonin may have both immediate and prospective mechanisms of action. As a result, during the dark phase melatonin triggers cellular and molecular mechanisms that are expressed only after cessation of the melatonin signal during the following day. Recent studies have reported that biorhythm disorders are closely associated with BC incidence and progression [69]. For example, the circadian gene *TIMELESS* was found to be highly expressed in BC conditions, predicting a poor prognosis (e.g., tumor growth) after positive regulation of sphingolipid metabolism [34].

To rapidly increase tumor mass, cancer cells intensify glucose uptake by utilizing aerobic glycolysis, termed the “Warburg effect”, rather than by oxidative phosphorylation [28,70]. In this context, previous studies have demonstrated that melatonin suppresses BC aerobic metabolism and consequently cell signaling pathways critical to cell survival, proliferation, metastasis, and drug resistance (reviewed by Hill et al. [42]). According to Blask and co-workers [38], increased levels of nocturnal melatonin hamper the Warburg effect and inhibit growth of BC. Melatonin has been shown to reduce the expression of the glucose transporter (GLUT-1), thereby limiting cell proliferation while inducing apoptosis in BC cell lines [71,72]. Similar to our results with hexoses (including glucose), in a BC xenograft model under light-dark 12:12 conditions, Blask et al. [38] observed a daily rhythmic pattern of tumor glucose uptake which was increased during the light phase and profoundly reduced in the dark phase. In comparison, we further showed that melatonin was effective in promoting a marked reduction in the levels of circulating hexoses during the peak at the 15:00 h, significantly sustaining its lower levels until the early dark phase. Melatonin may act via hypoxia-inducible factor-1 α (HIF-α), which in turn inhibits mitochondrial pyruvate dehydrogenase (PDH) during glycolytic metabolism, limiting the use of pyruvate-derived glycolytic carbons in the mitochondrial tricarboxylic acid cycle for cellular biosynthesis [38,73,74].

## 4. Materials and Methods

### 4.1. Cell Culture and Reagents

Melatonin and fetal bovine serum (FBS) were purchased from Sigma-Aldrich (St. Louis, MO, USA). The triple negative breast cancer (TNBC) cell line MDA-MB-231 was purchased from the ATCC (Manassas, VA, USA). Cells were grown in 75 cm^2^ flasks (Sarstedt, Nümbrecht, Germany) with DMEM (Life Technologies, Carlsbad, CA, USA) supplemented with 10% FBS (Cultilab, Campinas, SP, Brazil), penicillin (100 U/mL), and streptomycin (100 mg/mL) (Sigma-Aldrich, St. Louis, MO, USA). Culture medium was frequently replaced and cells were kept in a humidified chamber at 5.0% CO_2_ and 37 °C until they reached 80–90% confluence.

### 4.2. Human Breast Cancer Xenograft Model and Experimental Groups

A total of 160 Balb/c nude female mice, 7–8 weeks of age and weighing 16–18 g, were purchased from the Central Biotherium of the University of São Paulo/USP (São Paulo, Brazil). Animals were kept in pathogen-free conditions, at 23 °C and exposed to a normal diurnal variation under 12 h of light and 12 h of dark (lights on at 06:00 h and lights off at 18:00 h), with food and water available ad libitum. MDA-MB-231 human BC cells were grown, harvested, re-suspended in serum free media at a concentration of 6 × 10^7^ cells per mL, and inoculated into mice according to Junior et al. [35]. Mice (8 weeks old) were subcutaneously inoculated with 100 µL of MDA-MB-231 human BC cells directly in the fat pad (right mammary gland) and randomly divided into four experimental groups (*n* = 40/group): non-tumor bearing mice that received vehicle only (NTV), non-tumor bearing mice treated with melatonin (NTM), tumor-bearing mice that received vehicle (TV), and tumor-bearing mice treated with melatonin (TM) (Figure 10). These BC cells are efficient (~90%) in reproducing xenografted tumors (developed before 70 days of age) with remarkable vascularity and less central necrosis. All procedures were approved by the Ethics Committee on the Use of Animals of the Faculty of Medicine of São José do Rio Preto FAMERP (Protocol number: 001-003336/2014). The animal model and sampling were conducted in accordance with the ARRIVE guidelines and followed the relevant guidelines and regulations from the national and international standards for ethics in animal experimentation.

### 4.3. Melatonin Preparation and Treatment

Melatonin treatment was performed according to the method outlined by Jardim-Perassi et al. [39]. For the animals designated to receive melatonin (Sigma, St. Louis, MO, USA) as treatment, a 100 µL solution containing a dose of 40 mg/kg of body weight was administered by intraperitoneal injection (IP) for five days a week for 21 days, while non-treated mice (control animals) received IP injection with 100 µL of vehicle solution in a mixture ratio of 8 mL of phosphate buffered saline (PBS), 1 mL of dimethyl sulfoxide (DMSO), and 1 mL of Cremophor (Sigma, St. Louis, MO, USA). All treatments were given 1 h before lighting was switched off (at 17:00). The mice were periodically weighed and monitored during the experimental period and showed no variation in due course. Melatonin or vehicle solution was administered after cell implantation and tumor development for 21 days.

### 4.4. Plasma Sampling for Targeted Metabolomic Analysis

On the 22nd day after the initial treatment, mice were anesthetized with ketamine–xylazine and euthanized according to Junior et al. [35] at eight time points every three hours over a period of 24 h (06:00 h, 09:00 h, 12:00 h, 15:00 h, 18:00 h, 21:00 h, 00:00 h, and 03:00 h). During the dark phase, sampling was performed in systematic and standardized steps by avoiding artificial light in the collection room. We used a red dim light to illuminate the room during animal blood sampling on the bench. Blood samples were collected by open cardiac puncture at eight time points using a heparinized syringe, and plasma was obtained by centrifugation at ~5000 RCF for 2 min at 4 °C. The supernatant was recovered and frozen at −80 °C until subsequent analysis. Because the sampling frequency is usually 4–6 h for most chronobiological studies, we adopted more time points for collection over a longer duration. 

### 4.5. Targeted Metabolomic Analysis

Metabolites were measured using the AbsoluteIDQ p180 targeted metabolomics kit (Biocrates Life Sciences AG, Innsbruck, Austria), which covers six classes of metabolites (including amino acids, biogenic amines, acylcarnitines, phosphatidylcholines, lysophosphatidylcholines, hexoses, and sphingomyelins), on a Waters Xevo TQ-S mass spectrometer coupled to an Acquity H-Class LC system (Waters Corporation, Milford, MA, USA). 

### 4.6. Data Analysis and Statistics

All data were analyzed and processed using MassLynx V4.1 and validated by MetIDQ software (Biocrates Life Sciences AG, Innsbruck, Austria). Two-way analysis of variance (for two independent factors: tumor and treatment) complemented with Tukey’s test or Student’s t-test was performed using GraphPad Prism v. 9.0. The statistical significance was set at *p* value < 0.05 from the rhythmic analysis for every single metabolite among the groups in each time point. We used MetaboAnalyst 5.0 database (https://www.metaboanalyst.ca/) [75] for enrichment pathway evaluation (accessed on 20 July 2022). The heatmaps used for clustering analyses were performed using the web tool Morpheus (https://software.broadinstitute.org/morpheus) (accessed on 10 May 2022) [76].

## 5. Conclusions

Collectively, this report highlights the role of pharmacological doses of melatonin, a circadian-secreted molecule with anti-breast cancer effects, in controlling the plasma metabolite levels in BC-bearing mice. Furthermore, by sampling over 24 h we were able to accurately monitor tumor-associated changes in circulating metabolites through a complete light:dark cycle, thereby identifying key time-points for the actions of melatonin. Our results revealed a significant reduction in blood metabolite concentrations in the light and dark phases by melatonin, suggesting that this indoleamine may negatively affect BC bioenergetics and in turn alleviate the burden of breast cancer, supporting the use of melatonin as an adjuvant therapy. Melatonin administered at night is most effective as an anti-BC agent, as important metabolites associated with tumor bioenergetics are little available during daytime. Additional studies are needed to reevaluate this effect in other human tumor xenograft models following distinct genetic backgrounds. The utilization of syngeneic mouse models with very low genetic and metabolic differences could offer valuable insights in order to properly investigate the biological rhythms in cancer-related studies.

## Figures and Tables

**Figure 1 ijms-23-09105-f001:**
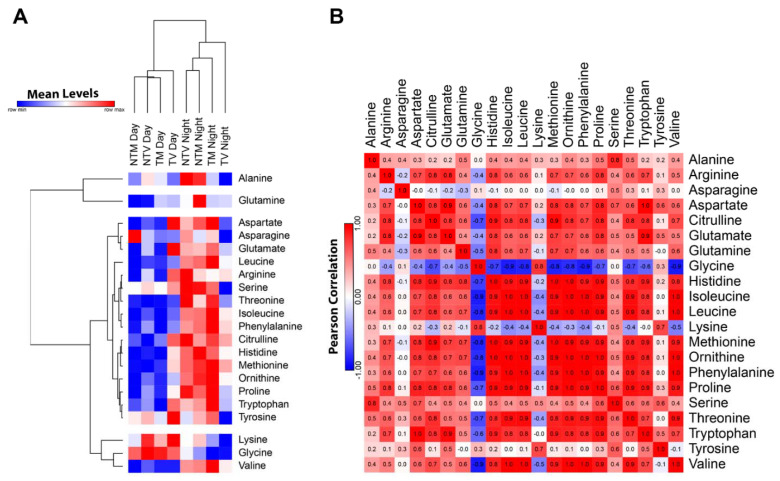
Profile of amino acid concentrations measured daily in tumor-bearing and non-tumor bearing mice that received melatonin as treatment. (**A**) Heat map illustrating the normalized mean levels of amino acids, visualized as minimum “blue” and maximum “red” values. We used two factors of clusterization (amino acids by row and experimental groups by column; Euclidean distance analysis). The heat map indicates the four main clusters of amino acids in the light and dark phases. Light phases included 06:00, 09:00, 12:00, and 15:00. Dark phase included 18:00, 21:00, 00:00, and 03:00 h. (**B**) Correlation matrix among plasma amino acids in the four experimental groups over 24 h. Pearson correlation shows negative and positive values.

**Figure 2 ijms-23-09105-f002:**
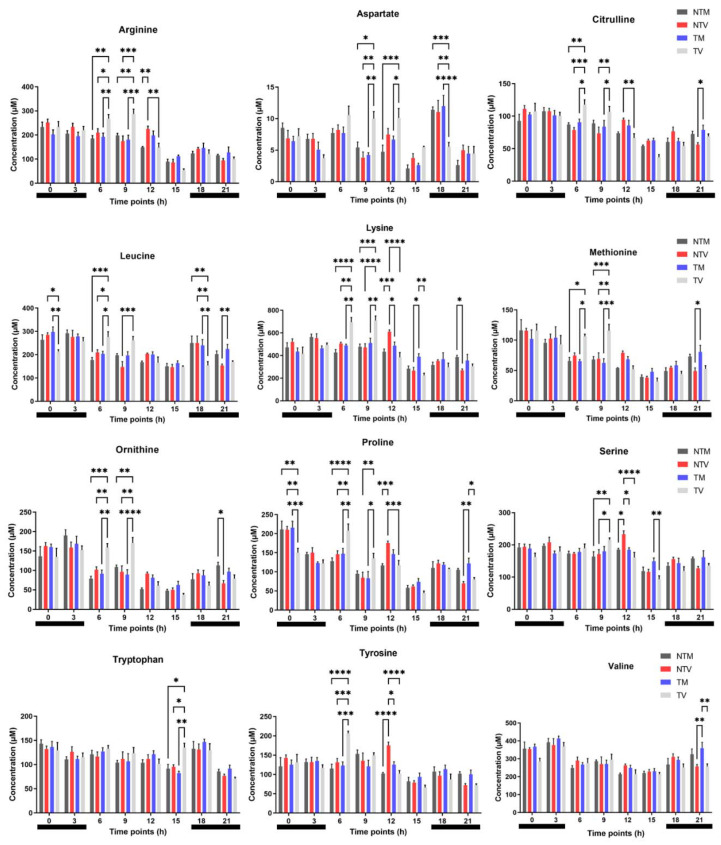
Concentrations of plasma amino acids in BC-bearing mice. The amino acids were measured at eight time points after treatment with melatonin or vehicle only. Light phase included 06:00, 09:00, 12:00, and 15:00 and dark phase included 18:00, 21:00, 00:00, and 03:00 (horizontal black bars). Data represent the mean ± SD of the concentrations (µM) among the experimental groups (NTM: non-tumor treated with melatonin; NTV: non-tumor treated with vehicle; TM: tumor treated with melatonin; TV: tumor treated with vehicle). Two-way ANOVA complemented with Tukey’s multiple comparisons test. * *p* < 0.05, ** *p* < 0.01, *** *p* < 0.001, **** *p* < 0.0001.

**Figure 3 ijms-23-09105-f003:**
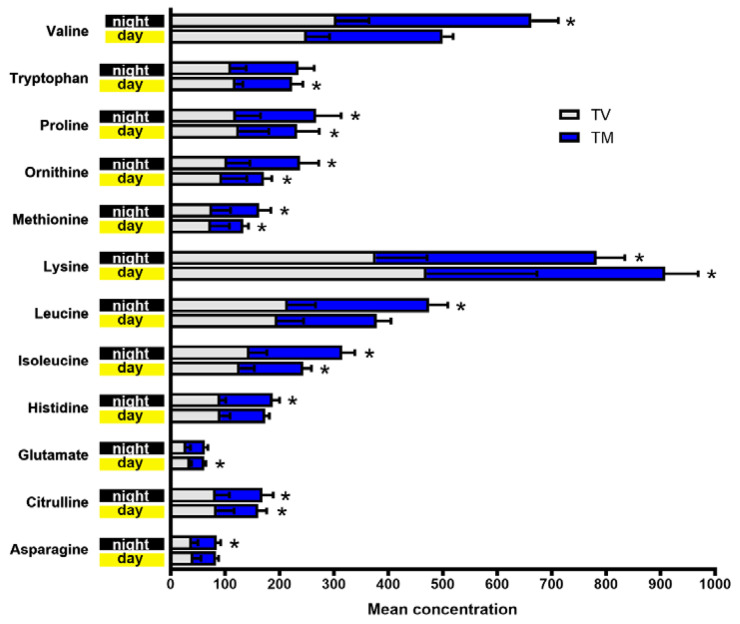
Changes in amino acid levels after melatonin treatment. Evaluation of 12 metabolites measured throughout the day and night in BC-bearing mice. TM: tumor treated with melatonin “blue horizontal bar”; TV: tumor treated with vehicle “soft grey horizontal bar”. Student’s *t*-test. * *p* < 0.05.

**Figure 4 ijms-23-09105-f004:**
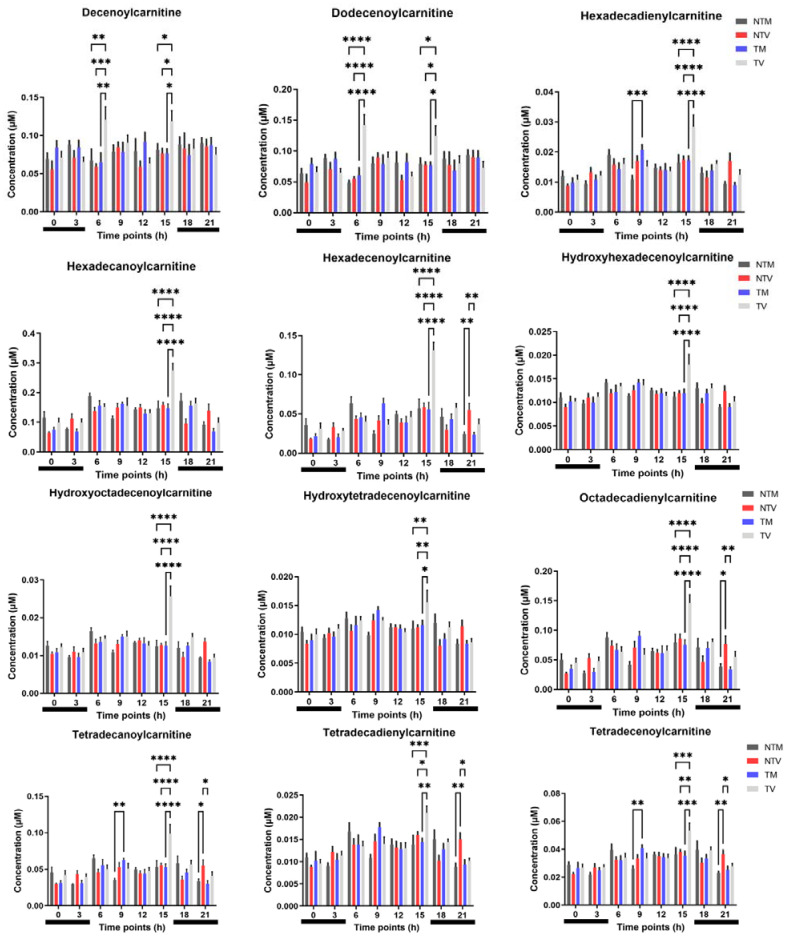
Concentration of plasma acylcarnitines in BC-bearing mice. Acylcarnitines were measured at eight time points after treatment with melatonin or vehicle only. Light phases included 06:00, 09:00, 12:00, and 15:00. Dark phases included 18:00, 21:00, 00:00, and 03:00 (horizontal black bars). Data represent the mean ± SD of the concentrations (µM) among the experimental groups (NTM: non-tumor treated with melatonin; NTV: non-tumor treated with vehicle; TM: tumor treated with melatonin; TV: tumor treated with vehicle). Two-way ANOVA complemented with Tukey’s multiple comparisons test. * *p* < 0.05, ** *p* < 0.01, *** *p* < 0.001, **** *p* < 0.0001.

**Figure 5 ijms-23-09105-f005:**
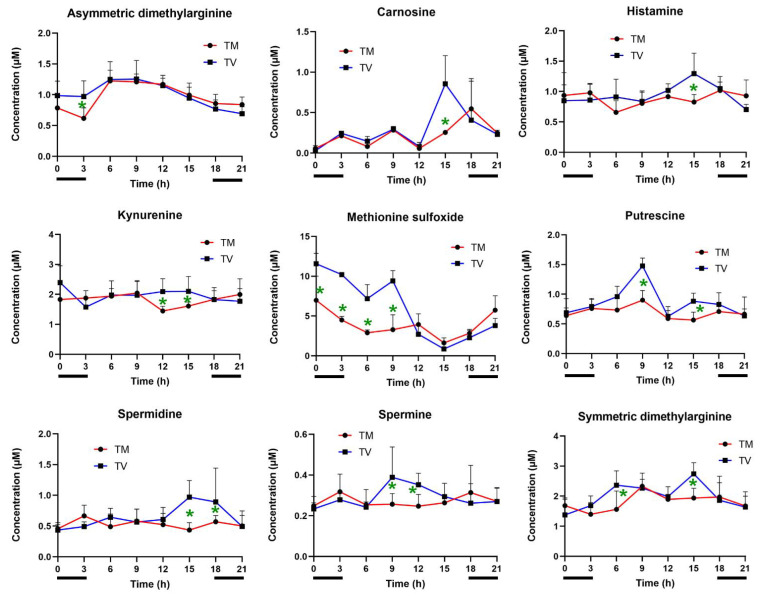
Concentration of biogenic amines in BC-bearing mice. Plasma biogenic amines were measured at eight time points after treatment with melatonin or vehicle only. Light phases included 06:00, 09:00, 12:00, and 15:00. Dark phases included 18:00, 21:00, 00:00, and 03:00 (horizontal black bars). Data represent the mean ± SD of the concentrations (µM) among the experimental groups (TM: tumor treated with melatonin; TV: tumor treated with vehicle). Student’s *t*-test. * *p* < 0.05.

**Figure 6 ijms-23-09105-f006:**
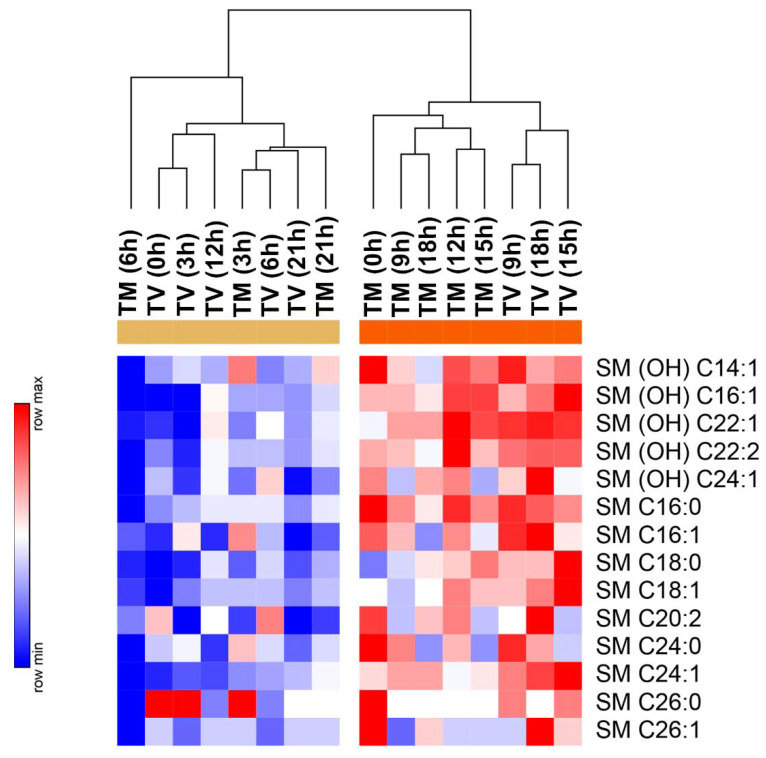
Profile of sphingolipids evaluated daily in BC-bearing mice after melatonin treatment. Heat maps illustrating the normalized mean levels of 14 different sphingomyelins, visualized as minimum “blue” and maximum “red” values. We used one factor of clusterization (melatonin treatment vs. vehicle throughout the day (06:00, 09:00, 12:00, and 15:00) and throughout the night (18:00, 21:00, 00:00, and 03:00) by columns; Euclidean distance analysis).

**Figure 7 ijms-23-09105-f007:**
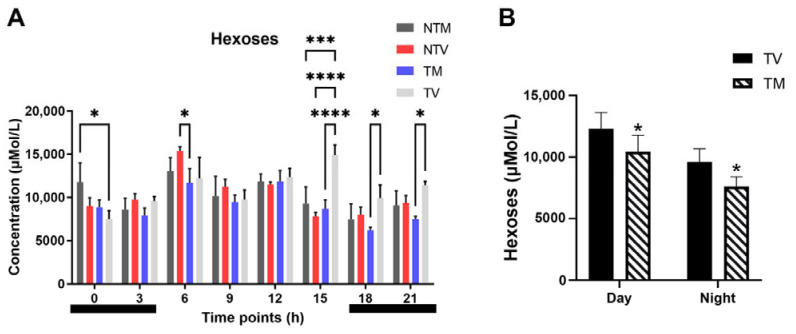
Variation in hexoses (including glucose) evaluated daily in BC-bearing mice after melatonin treatment. (**A**) Plasma levels of hexoses (µMol/L) at the eight time points among the experimental groups (NTM: non-tumor treated with melatonin; NTV: non-tumor treated with vehicle; TM: tumor treated with melatonin; TV: tumor treated with vehicle). Two-way ANOVA complemented with Tukey’s multiple comparisons test. * *p* < 0.05, *** *p* < 0.001, **** *p* < 0.0001. (**B**) Total concentration of hexoses throughout the day and night periods in BC-bearing mice treated with melatonin or receiving vehicle only. Data are expressed as mean ± SD. Student’s t-test. * *p* < 0.05.

**Figure 8 ijms-23-09105-f008:**
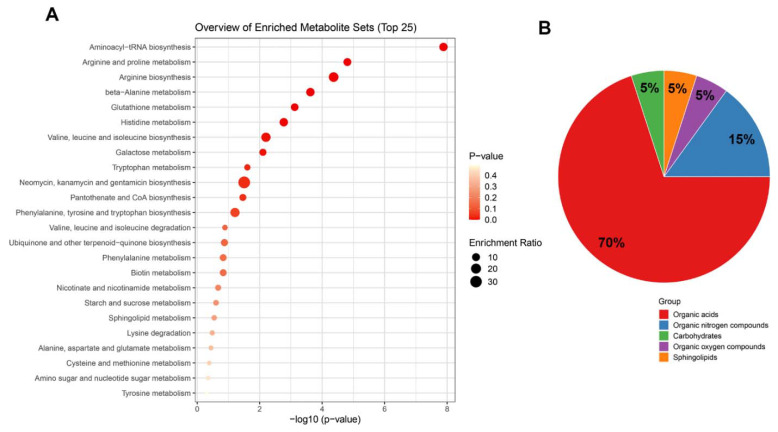
Top enrichment analysis of the metabolites modified by melatonin. (**A**) All metabolites were used to enrich terms based on KEGG metabolic pathways. (**B**) Pie chart showing the most representative chemical class metabolite sets. Analysis was performed in MetaboAnalyst v.5.0 using the metabolites within the Human Metabolome Database HMDB, and showed significant alteration in many pathways in a triple negative MD-MBA-231 breast cancer xenograft model.

**Figure 9 ijms-23-09105-f009:**
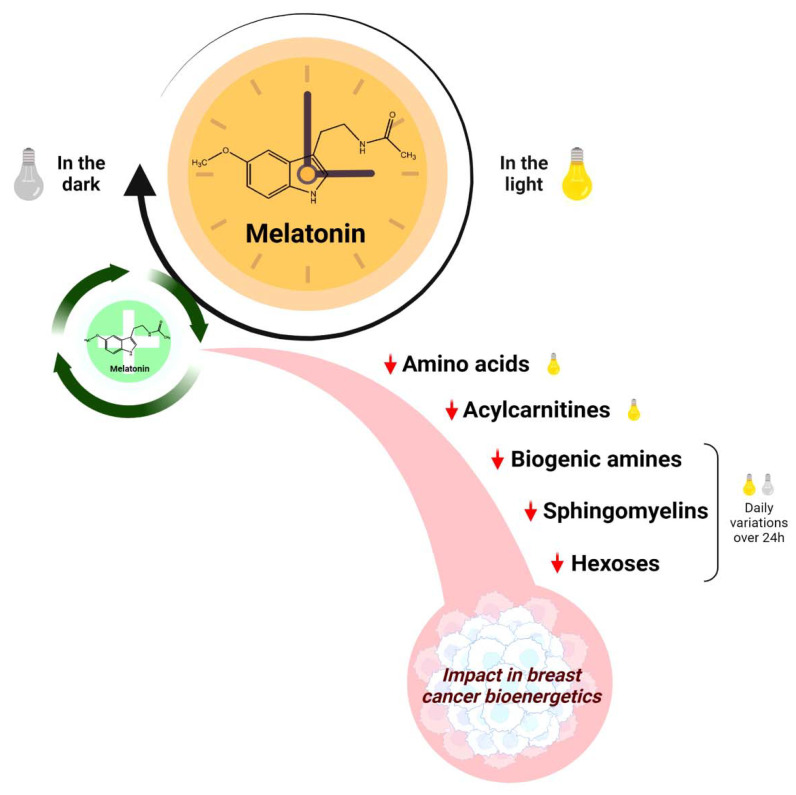
Melatonin has a controlled biosynthesis by the circadian clock (black circular arrow), with a significant decline in the light phase of the day (yellow lamp). A prolonged cycle of melatonin treatment in the early dark phase (green circular arrow) reduces the plasma concentrations of specific amino acids, acylcarnitines, biogenic amines, sphingomyelins, and hexoses mainly in the light phase of the day, which may impact breast cancer bioenergetics. Red arrows = reduced levels.

**Figure 10 ijms-23-09105-f010:**
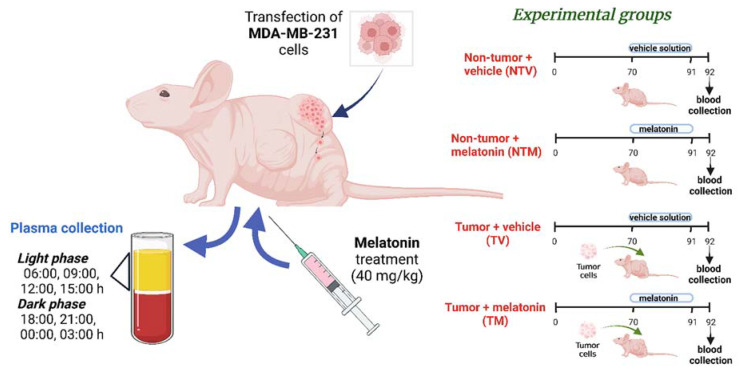
Representative illustration of the experimental procedures. After MDA-MB-231 cells were subcutaneously inoculated in female nude mice, the xenografted BC became apparent within a few days and four experimental groups were formed. Melatonin or vehicle solution was administered by intraperitoneal injections over 21 days. At 92 days of age, blood collection was carried out following eight specific time points. NTV: non-tumor animals treated with vehicle, NTM: non-tumor animals treated with melatonin, TV: tumor animals treated with vehicle, and TM: tumor animals treated with melatonin.

## Data Availability

All data generated or analyzed during this study are included in this article.

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
