# Peer review of "Melatonin Regulates the Daily Levels of Plasma Amino Acids, Acylcarnitines, Biogenic Amines, Sphingomyelins, and Hexoses in a Xenograft Model of Triple Negative Breast Cancer"

_ijms, 2022, doi:10.3390/ijms23169105_

Round 1
Reviewer 1 Report
The authors did a great job in finding out the change in biochemicals due to the melatonin treatment. Using of day/light criteria and the inclusion of in vivo model strengthened the manuscript. However, the reviewer felt that certain aspects can be improved. Please see the comments below.
1. Please carefully check for typos and grammatical errors. For example, there is a typo in "clinical study" on line 90.
2. The title can be improved. The current title is confusing since there is a lot of literature about the metabolism of melatonin itself. The inclusion of the major metabolites and the breast cancer model/type may help.
3. A schematic (or visual abstract) of the metabolites or major findings will enhance the readability.
4. The striking weakness of this study is the lack of functional assay. The reviewer did not understand the meaning of "the suppressive effects of melatonin on the proliferation of MDA-MB-231 cells are inconclusive in our model" on line 479. Please include any functional assay data (cell viability/migration, clonogenic assay, tumor volume, biomarkers such as ki67, p53, or proteins) if available. Even simple cell viability data will improve the study.
5. One way to support the mentioned inconclusiveness is by including a detailed mechanism of melatonin. Furthermore, melatonin has been used in combination/conjugation with other drugs to improve the efficacy in TNBC (for example, https://doi.org/10.1177/1178223420924634, https://doi.org/10.3390/cancers13133263, https://doi.org/10.1124/mol.119.116202). The inclusion of this finding will help in explaining the different anti-cancer effects of melatonin. Perhaps, other hormones play a significant role in melatonin's effect; that is why outcomes of melatonin treatment can vary.
Author Response
The authors did a great job in finding out the change in biochemicals due to the melatonin
treatment. Using of day/light criteria and the inclusion of in vivo model strengthened the manuscript. However, the reviewer felt that certain aspects can be improved. Please see the comments below.
Response: We really appreciate all of the points/issues raised by the reviewer in order to improve the quality and scientific value of the manuscript. We did our best to address these important issues.
- Please carefully check for typos and grammatical errors. For example, there is a typo in "clinical study" on line 90.
Response: As requested, english grammar and typos were carefully checked and corrected by a native speaker.
- The title can be improved. The current title is confusing since there is a lot of literature about the metabolism of melatonin itself. The inclusion of the major metabolites and the breast cancer model/type may help.
Response: Thank you for raising this issue. We agree with the reviewer. As requested, we modified the title by incorporating the major metabolites and breast cancer model.
- A schematic (or visual abstract) of the metabolites or major findings will enhance the readability.
Response: Thank you for raising this issue. We have created a schematic illustration of the main findings to improve the understanding of the manuscript (please see figure 9).
- The striking weakness of this study is the lack of functional assay. The reviewer did not understand the meaning of "the suppressive effects of melatonin on the proliferation of MDA-MB-231 cells are inconclusive in our model". Please include any functional assay data (cell viability/migration, clonogenic assay, tumor volume, biomarkers such as ki67, p53, or proteins) if available. Even simple cell viability data will improve the study.
Response: Thank you for raising this issue. In fact, we did not find a noticeable difference in tumor size after melatonin treatment in our xenograft tumor. However, melatonin at high concentration was already documented to inhibit proliferation of MDA-MB-231 cell line. In vivo and in vitro analysis involving melatonin’s effects in TNBC have been shown by our group and is mentioned in the new sentence (a number of evidences have been addressed in previous published studies). This may help the readers to understand the responsiveness of melatonin considering the subtypes of BC (please see discussion section on page 13).
- One way to support the mentioned inconclusiveness is by including a detailed mechanism of melatonin. Furthermore, melatonin has been used in combination/conjugation with other drugs to improve the efficacy in TNBC (for example,
https://doi.org/10.1177/1178223420924634, https://doi.org/10.3390/cancers13133263, https://doi.org/10.1124/mol.119.116202). The inclusion of this finding will help in explaining the different anticancer effects of melatonin. Perhaps, other hormones play a significant role in melatonin's effect; that is why outcomes of melatonin treatment can vary.
Response: Thank you for sharing these studies. As suggested, we have included these papers to strenghthen our findings based on the mechanism of melatonin in association with other hormones / agents to improve its efficacy in TNBC (please see page 13-14, lines 517-523).

Reviewer 2 Report
Junior RP et al. present a study on the role of melatonin in the profile of amino acids and metabolites present in the plasma of mice bearing human breast cancer xenografts (MDA-MB-231 cells). The authors evidence the effects of melatonin administration against breast cancer, supporting the use of melatonin as adjuvant therapy. Therefore, the present study highlights the need to expand the investigation of the biological rhythms in relation to cancer development.
Author Response
Junior RP et al. present a study on the role of melatonin in the profile of amino acids and metabolites present in the plasma of mice bearing human breast cancer xenografts (MDA-MB-231 cells). The authors evidence the effects of melatonin administration against breast cancer, supporting the use of melatonin as adjuvant therapy. Therefore, the present study highlights the need to expand the investigation of the biological rhythms in relation to cancer development.
Response: We are grateful to the reviewer comments regarding the value of our study. As requested, english grammar and style were carefully checked by a native speaker.
Reviewer 3 Report
The manuscript provides interesting results on the effect of melatonin in plasma metabolite levels in mice bearing xenograft triple negative breast cancer. Although it is an interesting work, there are a number of issues that precludes the publication of this paper in its present form. Below are some points that the authors should consider when reviewing their manuscript:
English language and style spell check is required
Figure 2: How do the authors interpret the fact that most measured AAs are significantly increased in NTV, compared to the other groups, in the 12h timepoint?
It would help the reader if the authors can include PCA score plot figures of the metabolic profiles of the different groups of samples, and the comparison of each group (e.g. TV vs TM; NTV vs TV; etc)
The authors should consider organizing the manuscript sections based on the groups of samples that are compared, including all types of metabolites analysed in each sample. It would help the reader to interpret the results.
Would the authors explain why do they perform a different top enrichment analysis for each metabolite group? Could they include the top enrichment analysis results for the analysis of all the metabolites measured in the samples?
Author Response
The manuscript provides interesting results on the effect of melatonin in plasma metabolite levels in mice bearing xenograft triple negative breast cancer. Although it is an interesting work, there are a number of issues that precludes the publication of this paper in its present form. Below are some points that the authors should consider when reviewing their manuscript:
Response: We really appreciate all of the points/issues raised by the reviewer in order to improve the quality and scientific value of the manuscript. We did our best to address these importante issues.
English language and style spell check is required
Response: As requested, english grammar and style were carefully checked by a native speaker.
Figure 2: How do the authors interpret the fact that most measured AAs are significantly increased in NTV, compared to the other groups, in the 12h timepoint?
Response: Thank you for the appropriate observation by the reviewer. In fact, the NTV group has the highest amounts of AAs at 12 h, and this can be interpreted by the diet metabolism which likely promoted a greater bioavailability of AAs in the tumor-free animals. A brief comment regarding this aspect has been added to the discussion section (please see on page 13, lines 510-513).
It would help the reader if the authors can include PCA score plot figures of the metabolic profiles of the different groups of samples, and the comparison of each group (e.g. TV vs TM; NTV vs TV; etc)
Response: Thank you for raising this issue. We discussed on the possibility of including PCA plots, but it would render a number of additional data based on the eight time points, four experimental groups, and specific metabolites. In fact, PCA scores are very helpful for the readers. Ee have already created PCA models for each sample and group in previous analyses; however they were not suggested by most of reviewers because of the color overlay on the graphs. The number of time points generate eight different colors in each model that are very difficult for the visual sample separation. Thus, it would be even more confusing for the reader understanding. In this case, we believe that directly showing the heat maps and bars graphs could generate better understandings.
The authors should consider organizing the manuscript sections based on the groups of samples that are compared. It would help the reader to interpret the results.
Response: Thank you for raising this issue. As requested, we re-organized the results section considering the specific groups of samples and metabolites that are presented in the topic. Some figures were separated to maintain this organization.
Would the authors explain why do they perform a different top enrichment analysis for each metabolite group? Could they include the top enrichment analysis results for the analysis of all the metabolites measured in the samples?
Response: Thank you for raising this issue. We agree with the reviewer. As requested, we provided a new top enrichment analysis including all the metabolites measured in the samples (please see Figure 8).
Round 2
Reviewer 1 Report
The manuscript is in good condition. However, there are some points for improvement that still exist.
1. Please include the percent value on the pie chart for figure 8.
2. Please make sure the figures are in high resolution. The resolutions on the PDF were not good enough.
3. The reviewer felt figure 9 could be improved a lot. Please include more details if possible such as defining circle arrows, including specific biomarkers and their function, and how melatonin/cancer regulates that biomarker. A cell structure can be added. Furthermore, biomarkers-related receptor activities (melatonin and others) can be added.
Author Response
The manuscript is in good condition. However, there are some points for improvement that still exist.
Response: We really appreciate all of the points/issues raised by the reviewer in order to improve the quality and scientific value of the manuscript. We did our best to address these important issues.
1. Please include the percent value on the pie chart for figure 8.
Response: Thank you for raising this issue. As requested, we modified figure 8 by adding the percent of individuals metabolites classes.
2. Please make sure the figures are in high resolution. The resolutions on the PDF were not good enough.
Response: Thank you for raising this issue. All figures are at high resolution (higher than 300 dpi). The pdf conversion indeed reduces the quality/resolution of the images.
3. The reviewer felt figure 9 could be improved a lot. Please include more details if possible such as defining circle arrows, including specific biomarkers and their function, and how melatonin/cancer regulates that biomarker. A cell structure can be added. Furthermore, biomarkers-related receptor activities (melatonin and others) can be added.
Response: Thank you for sharing this point. We agree that it would be more attractive a schematic figure illustrating the intracellular mechanisms by which melatonin regulates the cell metabolism. However, we only detected the metabolites in the plasma, and we could not infer how these molecules could be interacting in the breast cancer cell, neither how is the mechanism, cell sites, melatonin regulation, etc. These analyzes will be deeper investigated in future studies to safely understand its actions into the tumor cell. Because we can’t assure about the mechanims of regulation, we kept the previous image and improved the symbols and descriptions as recommended. We are open to hear from the reviewer considering other ways to improve the image.